# Ocular Barriers and Their Influence on Gene Therapy Products Delivery

**DOI:** 10.3390/pharmaceutics14050998

**Published:** 2022-05-06

**Authors:** Bastien Leclercq, Dan Mejlachowicz, Francine Behar-Cohen

**Affiliations:** 1Centre de Recherche des Cordeliers, From Physiopathology of Ocular Diseases to Clinical Development, Sorbonne University, Université de Paris Cité, Inserm, F-75006 Paris, France; bastien.leclercq@inserm.fr (B.L.); mejlachowicz.dan@hotmail.fr (D.M.); 2Assistance Publique Hôpitaux de Paris, Ophtalmopole, Cochin Hospital, Université de Paris Cité, F-75015 Paris, France; 3Department of Ophthalmology, Hôpital Foch, F-92150 Suresnes, France

**Keywords:** eye, barriers, retina, cornea, drug delivery, bioavailability

## Abstract

The eye is formed by tissues and cavities that contain liquids whose compositions are highly regulated to ensure their optical properties and their immune and metabolic functions. The integrity of the ocular barriers, composed of different elements that work in a coordinated fashion, is essential to maintain the ocular homeostasis. Specialized junctions between the cells of different tissues have specific features which guarantee sealing properties and selectively control the passage of drugs from the circulation or the outside into the tissues and within the different ocular compartments. Tissues structure also constitute selective obstacles and pathways for various molecules. Specific transporters control the passage of water, ions, and macromolecules, whilst efflux pumps reject and eliminate toxins, metabolites, or drugs. Ocular barriers, thus, limit the bioavailability of gene therapy products in ocular tissues and cells depending on the route chosen for their administration. On the other hand, ocular barriers allow a real local treatment, with limited systemic side-effects. Understanding the different barriers that limit the accessibility of different types of gene therapy products to the different target cells is a prerequisite for the development of efficient gene delivery systems. This review summarizes actual knowledge on the different ocular barriers that limit the penetration and distribution of gene therapy products using different routes of administration, and it provides a general overview of various methods used to bypass the ocular barriers.

## 1. General Concepts

The eye is an organ isolated from the rest of the body and from the external environment by different ocular barriers. These barriers are the result of structural, dynamic, and functional elements that protect the eye from the outside and from systemic circulation. They are formed during embryonic development, and their integrity and functionality can be modified in pathologies and during aging. Barriers are essential to protect visual structures from mechanical, chemical, metabolic, and pathogen damages. They are dependent on the activity of active and controlled transporters which preserve the transparency of tissues and ocular environments. This activity is essential to maintain the composition of ocular compartments and sometimes even intratissue microenvironments.

One of the major roles of barriers is to preserve the immune privilege of different ocular compartments such as the cornea, the aqueous humor, or the subretinal space. While barriers guarantee the integrity and the functions of the ocular tissues, they also constitute an obstacle to the penetration of gene therapy products. Therefore, they represent an intervention target for the development of formulations, vectors, and devices aimed at increasing the bioavailability of gene therapy products in the ocular compartments and gene delivery to the targeted cells. However, the barriers significantly limit the systemic passage of molecules and vectors administered into the eyeball, allowing true local administration and reduced immune reaction.

A better understanding of the various ocular barriers is a prerequisite for the development of gene therapy products designed to target specific tissues and/or cells. In this paper, we summarize the current knowledge on ocular barriers of the anterior and posterior segments of the eye with particular emphasis on aspects related to nucleic acids and gene delivery, using viral or nonviral vectors, rather than immune aspects, which can be consulted in an excellent recent review [1]. The description is provided from the outside to the inside, separating the anterior segment from the posterior segment, since the galenic forms used depend on the targeted segment. We restrict the review to the description of the ocular barriers and the known size and molecular weight limits of the different tissue and cell structures, as well as to the impact of such barriers on the development of gene therapy products for the treatment of ocular diseases.

## 2. Molecular Components of Tissue Barriers

Tissue barriers result from junctions between the cells that compose them, representing a major structural element. Furthermore, they also result from the superposition or assembly of structures with specific physicochemical characteristic that form obstacles to the penetration of certain compounds (lipid bilayer, molecular filters formed by collagen fibrils, electrostatic repulsion, etc.). Molecular films of glycosaminoglycans called the glycocalyx, along with the tear film, also contribute to the tissue barriers. Lastly, efflux proteins play a crucial role in the function of tissue barriers. The innate and adaptive immune systems, as well as the autonomic nervous system, particularly through its sensory and mechanical innervations, also regulate the properties and function of tissue barriers. The barriers should be considered as a dynamic, constantly moving system that receives signals from the microenvironment and expends energy to maintain its integrity. Here, we briefly summarize the components of cell junctions and efflux proteins. For more detail, the reader may refer to specialized journals.

### 2.1. Junction Proteins

Transmembrane proteins

The claudin family (1 to 27) comprises tetraspanic transmembrane proteins that play a central role in the control of paracellular transport, notably by forming ion-selective barriers and pores. Claudins connect to the actin cytoskeleton by interacting with the PDZ domain of ZO-1.

In the retina, vascular endothelial cells express claudins 1, 2, and 5. Claudins 3, 4, and 23 are found within the ganglion cell layer of the retina, while claudins 4 and 12 are present in the outer plexiform layer. Lastly, claudin 23 is expressed in the inner nuclear layer. Their specific roles are not yet clear [2].

MARVEL domain-containing proteins (MAL and related proteins for vesicle formation and membrane link) associated with tight junctions includes occludin, tricellulin, and MARVELD3 proteins. These proteins all contain a MARVEL domain with four conserved transmembrane domains. Occludin (or MARVELD1) is a protein of about 60 kDa, whose N-terminus cytoplasmic end interacts with partners that regulate its degradation and endocytosis, while the C-terminus binds zonula occludens proteins ZO-1, -2, and -3. Current research argues more for a modulatory role of occludin in the barrier and vesicular transport than a structural role in tight junctions. The membrane targeting of occludin and its functions are highly dependent on its phosphorylation by specific kinases [3].

The immunoglobulin superfamily includes the JAM (junction-associated molecules), the ESAM (endothelial selective adhesion molecule), and the CAR (coxsakie virus-adenovirus receptor) families.

JAMs comprise at least one IgG domain within their extracellular N-terminus, and their cytoplasmic end interacts with ZO-1 and the cytoskeleton. JAMs are implicated in junctional assembly facilitation, but not in the formation of tight junction strands. Colocalized with PAR-3 at intercellular junctions of the retinal pigment epithelium, JAM-A promotes localization of ZO-1 and occludin within cell–cell contact zones, resulting in barrier formation. JAMs contribute to several biological processes such as cell migration, infiltration of immune cells, and angiogenesis [4,5].

Cytoplasmic proteins

Proteins with a PDZ domain (ZO-1, -2, -3, MAGI-1, -2, -3, PALS-1, PATJ, MUPP-1, PAR-3, PAR-6, mDlg, Scrib, and afadin) are cytoplasmic scaffold proteins that play a central role in the organization of tight junctions, by connecting transmembrane proteins to the cytoskeleton. The lack of all ZO isoforms leads to an impairment in tight junction organization and claudin polymerization.

Other proteins

Other proteins include Cingulin, Symplekin, heterotrimeric G protein, PP2A, PTEN, Pilt, Rab3b, Rab13, aPKC, CRB3, LYRIC, CASK/LIN-2, Merlin, ZONAB, huASH1, GEF-H1, Angiomotin/JEAP, and TAZ/YAP.

### 2.2. The Tight Junctions (or Zonula Occludens)

In ultrafine cellular sections, tight junctions come into sight as close apposition contact points, also called “kissing points”, where the lipid bilayers of each cell are hardly distinguishable. These junctions are particularly located on the apical side of the polarized membrane. In endothelial barriers, several points of contact can be observed, located along the paracellular space (Figure 1A,B1).

Tight junctions ensure two major functions: the first is a gate function, by limiting the passage of molecules into the paracellular space; the second is a barrier function, which maintains the apical–basal polarity, by restraining the movement of molecules between the apical and the basolateral plasma membrane. However, tight junctions are also implicated in numerous cell signaling processes, e.g., cell proliferation, gene expression or differentiation.

At the molecular level, these junctions are composed of more than 40 types of proteins, classified into transmembrane proteins (occludin, claudin, JAMs, etc.) and cytoplasmic scaffold proteins (ZO, PAR, etc.) bound to the actin cytoskeleton. The correct localization of the various proteins, particularly their targeting to the membrane, depends on their state of phosphorylation.

### 2.3. The Adherens Junctions (or Zonula Adherens)

Under an electron microscope, these junctions appear as thickenings at the level of the plasma membrane. A higher magnification allows to visualize bridges between the membranes. These bridges are made up of transmembrane molecules, the cadherins, whose head contains a homophilic recognition system (Figure 1A,B2).

Adherens junctions play an essential role in cell–cell adhesion, apico-basal polarity, contact inhibition, and paracellular transport. Adherens junctions of the inner blood–retinal barrier (BRB) include cadherin (VE), a Ca^2+^-dependent transmembrane cell adhesion protein containing a well-conserved cytoplasmic tail which can bind to β-catenin.

### 2.4. The Desmosomes (or Macula Adherens)

Desmosomes are disc-shaped structures approximately 0.1 to 0.5 μm in diameter and 0.1 μm thick (Figure 1A,C3). They ensure intercellular bonds through transmembrane molecules of the cadherin superfamily. These molecules are related to the desmosomal plaque, which notably contains plakoglobin and desmoplakins.

### 2.5. Gap Junctions

The gap junction is an assembly of a few tens to a few thousand channels (connexons) that cross the two cell membranes, forming junctional plates. Each connexon is a hexamer of six transmembrane proteins (of the connexin type), forming a hydrophilic channel 2 nm in diameter (Figure 1A,D). This channel allows the passage of small soluble molecules (roughly corresponding to molecules of relative molecular weight less than 1200 Da) such as ions, amino acids, and some metabolites. In the lens, aquaporin 0 forms a very dense set of channels, which allow exchanges between the aqueous humor and the lens through epithelial cells (Figure 1D).

### 2.6. Selective Passage of Barriers under Physiological Conditions

Paracellular and transcellular transport across barriers involves six finely tuned mechanisms [6].

Passive paracellular diffusion: This is the only transport mechanism that does not require energy consumption because the tight junction physically limits the passage, e.g., to gases such as carbon dioxide or oxygen and to lipophilic molecules < 400 Da. Water does not cross these junctions.Facilitated cellular diffusion: Membrane transporters allow the passage of substances in solution following a concentration gradient. Glucose transport by glucose transporter 1 (GLUT1) is an example.Active transport: Membrane transporters hydrolyze ATP to displace dissolved substances against a concentration gradient or to create electrochemical gradients which lead to vector transport (e.g., Na^+^/K^+^ ATPase).Transcytosis: The cell membranes invaginate at the level of lipid rafts to form vesicles which can form true transient channels by fusion. Fusion with the opposite membrane releases the vesicular contents on the other side of the cell. Invaginations can be modulated by specific single or multi-ligand receptors. For instance, albumin crosses endothelial and pigment epithelial cells via a vesicular transport mechanism. The GP60 protein allows albumin binding to the membrane which invaginates and forms vesicles (or caveolae) mediated by caveolin-1.Modification of solutes: During their transport, molecules can be degraded or transformed. For example, retinol goes through the basal face of the retinal pigment epithelium via receptor-mediated endocytosis and is released into microsomes before being converted to *cis*-retinal. Then, *cis*-retinal crosses the apical monolayer and is endocytosed by the photoreceptors before binding to opsin. Another example is the CO^2^, which is first converted to HCO_3_^−^ before being transferred from the apical to the basal side of the cell monolayer.Efflux proteins: Efflux is a mechanism via which cells release intracellular compounds or compounds that have entered from outside the cell. It is an active, energy-dependent transport provided by efflux proteins responsible for the elimination of toxins, drugs, or chemicals. These pumps also regulate the passage of lipophilic molecules such as corticoids, cholesterol, phospholipids, or retinoids. Efflux mechanisms contribute to the emergence of resistance to various pharmacological treatments such as resistance to antibiotics or anticancer chemotherapy.

The P-glycoprotein is the main efflux pump in humans. It belongs to the ABC transporter family, which is subdivided into seven subfamilies of protein encoded by 49 genes. The efflux pumps act generally unidirectionally, from the cytoplasm to the extracellular space. Some pumps have essential physiological functions such as the excretion of toxins by the liver and restriction of penetration of toxic molecules into vital organs such as the brain or the eye. As such, efflux pumps are an integral part of eye barriers.

## 3. Description of Ocular Barriers

### 3.1. Barriers in the Anterior Segment

#### 3.1.1. The Eyelids and the Tear Film

The eyelids constitute a mechanical barrier to the penetration of bacterial agents, foreign bodies, and any external aggression. The reflex blink, controlled by the sensory innervation of the cornea, is a major factor in the elimination of instilled drops, since it is estimated that 10% to 20% of the instilled volume is eliminated by the blink. This estimation can be increased by the use of formulations with an acidic or basic pH, a factor of immediate poor tolerance. The tear film represents a barrier due to its physicochemical properties, as well as the antibacterial properties and the anti-inflammatory enzymes and proteins it contains. The tear film is formed by three different layers, the lipid layer (outer), which is produced by the meibomian glands, the aqueous layer (middle) which is secreted by the lacrimal glands, and the mucinic layer (inner) which is secreted by the mucous glands and glandular cells of the conjunctiva. The lipid layer limits the natural evaporation of tears and, thus, maintains their osmolarity (290–340 mOsm in normal conditions) [7].

The aqueous layer allows the humidification of the eye and the transport of oxygen and carbon dioxide. This layer contains the soluble substances which not only ensure anti-infectious and antioxidant defense (lysozyme, lactoferrin, defensins, IgG A, etc.) but also growth factors (EGF, NGF, etc.) essential to preserve the integrity of the epithelial barrier. Lactoferrin, found in large quantity in tears [8], is recognized as a strong protective agent against SARS-CoV-2, an agent of COVID-19 [9], which may partly explain the low penetration of the virus through the eye.

The deep mucin layer forms a slightly viscous film, which also participates in the barrier against pathogens by limiting their adhesion and promoting their elimination. The corneal and conjunctival epithelial cells are responsible for the production of membrane-associated mucins (MAM), notably MUC1, MUC4, and MUC16, which represent the main constituents of the glycocalyx. The goblet cells produce gelling mucin (essentially MUC5AC). This biofilm is essential for anti-infectious defense and helps to eliminate pathogens and foreign bodies. The normal volume of the tear film, 8–10 µL, also contains enzymes able to metabolize drugs (prodrugs) and/or to inactivate drugs [10].

#### 3.1.2. The Conjunctiva

Conjunctival cells are bound by tight junctions limiting the paracellular movements of large molecules and pathogens. These tight junctions are constituted by claudins (1, 2, 4, 7, 9, 10, and 14), occludin, JAM-A, and tricellulin. They are bound to the cytoskeleton by ZO-1, ZO-2, and ZO-3. However, the resistance of the conjunctival epithelium is low at about 1500 Ω·cm^2^. It is accepted that a molecular weight between 20 and 40 kDa (MW) is the limit for conjunctival penetration [7] (Figure 2). Nevertheless, the intraocular penetration of molecules through the conjunctiva is also limited by enzymatic degradation and by rapid vascular drainage. Following a subconjunctival injection, the albumin reaches the cervical lymph nodes in less than 6 min, and molecules of small molecular weight which cross the conjunctival barrier are, for the most part, eliminated even more quickly by vascular and lymphatic drainage, preventing them from reaching the posterior sclera or the retina [11]. Thus 40% to 80% of molecules diffuse via the systemic route [12]. Aging, hyperosmolarity (especially related to hyperglycemia), mucus reduction, and biofilm instability influence the penetration of drugs through the conjunctival barrier. Nevertheless, after topical instillation, some drugs are absorbed, at least in part, through the transconjunctival pathway such as atropine, latanoprost, carbonic anhydrase inhibitors, antihistaminic drugs, or insulin.

#### 3.1.3. The Cornea

The epithelial barrier constitutes 90% of the resistance to corneal penetration of molecules with a molecular weight > 500 Da. This barrier contains tight junctions made of occludin and claudins (2, 3, 4, 7, 9, and 14) at the apical pole of the superficial cells of the epithelium and of claudin 1 at the basolateral pole, leading to a high resistance of approximately 7500 Ω·cm^2^, comparable to the resistance in the intestine. The drugs can penetrate through the epithelium via three ways: the transcellular pathway, the paracellular pathway for hydrophilic molecules (<500 Da), and transporter-mediated permeation. Since the membranes of epithelial cells are negatively charged, molecules with a negative charge such as nucleic acids do not easily cross this layer.

The cornea has a “sandwich” structure, containing three different layers:lipid (epithelium),aqueous (stroma),lipid (endothelium).

This organization also constitute an obstacle to the penetration of hydrophilic (due to the epithelium) or hydrophobic (due to the stroma) molecules. Both Bowman’s membrane and Descemet’s membrane are not considered as barriers to drug penetration. A typical example is fluorescein, which remains on the surface of healthy epithelium but penetrates the stroma in the case of corneal erosion or alteration of the glycocalyx structure [13]. Lipophilic drugs can penetrate and be stored in the epithelium. Thus, only amphiphilic active ingredients, such as fluorometholone, and weak bases or acids in solution at a physiological pH pass into the aqueous humor through the cornea, such as pilocarpine or timolol. Apart from these two types of active ingredients which can be formulated in solution without preservatives, transcorneal absorption requires the addition of excipients that destabilize the corneal barrier or particulate formulations [14].

#### 3.1.4. The Iris and the Ciliary Body (the Hemato-Aqueous Barrier)

The blood–aqueous barrier consists of the nonpigmented epithelium, the ciliary body, and the capillary endothelium, which is watertight in the iris (Figure 2) because the ciliary body is vascularized by fenestrated and permeable capillaries. However, the iris capillaries contain tight junctions but have a higher permeability than the inner retinal capillaries due to intense trans-endothelial transport. At the level of the ciliary body, the cells of the pigmented epithelium are in contact with the capillaries of the choroid, while the basolateral membrane of the nonpigmented epithelium is in contact with the aqueous humor. Pigmented and nonpigmented cells communicate through gap junctions. While nonpigmented epithelial cells are joined by tight junctions, the latter are absent in pigmented cells. Thus, the physical barrier to the penetration of active principles through the ciliary body is the unpigmented epithelium (Figure 2B).

The concentration of protein in the aqueous humor (AH) is low, and only a fraction comes from the plasma. Indeed, plasma proteins that have crossed the fenestrated capillaries of the ciliary body cannot reach the posterior chamber as they are arrested by the unpigmented epithelium of the ciliary body, and the plasma proteins also cannot reach the anterior chamber because of the tight junctions of the iris capillary endothelium. However, plasma proteins that cross the fenestrated capillaries of the ciliary body might diffuse from the stroma of the ciliary body to the stroma of the iris and then reach the aqueous humor following concentration gradients. The concentration of smaller proteins in the aqueous humor reflects plasma concentrations, but there is no such correlation for large proteins, which diffuse much more slowly in the anterior chamber [15].

Limiting and controlling the penetration of drugs from the blood to the aqueous humor is not just about physical barriers. Indeed, the iris and ciliary body display a variety of drug transporters and efflux pumps. SLC (solute carrier) family drug transporters include organic anion transporters, bile acid, cation transporters, and peptide transporters. These drug transporters in the iris and ciliary body restrain the passage of drugs from the blood to the aqueous humor. Therefore, they reduce the bioavailability of drugs in the eye and also participate in the active removal of drugs from the aqueous humor [16].

### 3.2. The Sclera

The sclera constitutes chains of collagen and elastin that form a matrix of fibers whose pore diameter and intracellular space can define the circulation of macromolecules [17]. The average thickness of the human sclera is about 0.53 mm at the limbus, 0.39 mm at the equator, and 0.9–1.0 mm nearby the optic nerve. With an average total surface area of 16.3 cm^2^, the sclera provides an important gateway for drug delivery [18]. The scleral permeability to macromolecules decreases exponentially with the molecular weight of the drugs, but even proteins the size of an antibody can slowly penetrate through the sclera [19]. This effect is maximum at the equator and in the posterior superior temporal quadrant, allowing the passage of molecules up to 185 kDa and 9.67 nm [20,21]. The Einstein–Stockes radius (or charged molecular radius) seems to better predict scleral permeability compared to molecular weight only, with globular proteins being more permeable than linear dextrans of the same molecular weight. During aging, scleral thickness remains stable but the permeability of the human sclera decreases. This reduction is due to the crosslinking and glycation of collagen, which reduce scleral compliance and hydration [22], as well as increase rigidity, particularly in age-related macular degeneration [23]. The interfibrillar distance reduction primarily affects macromolecular permeability.

The effect of physical treatments such as hydration, cryotherapy, crosslinking, transscleral diode laser, and surgical thinning has been studied. In rabbits, the permeability to macromolecules increases with tissue hydration. In humans, surgical thinning increases the permeability to macromolecules. Cryotherapy and transscleral application of a diode laser had no significant impact [18,24], but crosslinking reduces scleral permeability up to 200 µm around the treated area [25].

Physiologically, choroidal proteins undergo very rapid transscleral elimination (less than 1 h) to the conjunctival lymphatics [26] or along the vascular network and nerves that pass through the choroid, which allows maintaining the oncotic gradient between the vitreous and the choroid, which lacks lymphatic vessels.

### 3.3. Impact of Ocular Barriers on Nucleic Acid and Gene Delivery to Tissues and Cells of the Anterior Segment of the Eye

Systemic administration of gene therapy products, such as viral vectors, naked nucleic acids or plasmid DNA, cell-penetrating peptides, nucleic acids encapsulated or conjugated with polymers, or cationic lipids, is not adapted for the treatment of diseases affecting the anterior segment of the eye, as they do not reach ocular tissues.

The most straightforward way to target the cornea is the topical instillation of naked nucleic acids, but the tear film contains enzymes that degrade the nucleic acids, while the negative charge of nucleic acids creates repulsive forces with the corneal epithelium. Indeed, the tears contain a high endonuclease activity, mostly ensured by lipocalin 2 [27], but low RNase activity [28]. The role of mucin in corneal drug bioavailability is not univocal [29], but muco-adhesion can be used to prolong the contact time of a formulation, as proposed for dexamethasone–glycol chitosan nanoparticles [30].

After instillation, naked oligonucleotides mostly reach superficial corneal and conjunctival cells in normal conditions but can be delivered to stromal cells or to pathological neovessels in pathologic conditions [31,32,33]. A naked oligonucleotide directed at insulin receptor substrate 1 (IRS-1) was evaluated for post-infectious corneal neovascularization using topical delivery [34]. More recently, topical instillation of a small interfering oligonucleotide of RNA (siRNA) silencing transient receptor potential vanilloid (TRPV1) was evaluated for the treatment of dry eye disease [35]. To reach deeper corneal layers, anterior segment tissues, and even limbal stem cells, the subconjunctival injection of naked nucleic acids interfering with RNA translation is efficient and allows reducing the frequency of administration [32]. Iontophoresis, through the creation of a transient low intensity electric field, allows the penetration of negatively charged nucleic acid molecules to the cornea and the anterior segment tissues [28,36]. Liposomal formulations also enhance the delivery of antisense oligonucleotides into corneal cells and in the iris [33]. Nanoparticulate systems, designed to favor ocular surface residency time by modulating their physicochemical properties, are being developed; however, if they enhance the epithelial delivery of nucleic acids or plasmid DNA, they still need to escape the endo-lysosomal compartment and avoid degradation of the biological cargo [37,38,39].

Due to the corneal barriers and the quick tear drainage, topical delivery of viral vectors can be efficient if corneal barriers are disrupted by diseases processes. Alternatively, the intrastromal administration of viral vectors and the targeted delivery using microneedles have been evaluated [40]. Plasmid DNA on the other hand can be delivered to cells in the anterior segment of the eye using physical methods such as current, laser, light, ultrasound, guns, or chemical methods, but it still needs to bypass the corneal barriers [40]. Naked plasmid DNA, injected in the ciliary muscle, has been efficiently transduced in ciliary muscle cells using a disposable electrotransfer device, allowing the sustained intraocular production of therapeutic proteins in the ocular media for more than 6 months [41,42,43,44,45].

In summary, delivering gene therapy products to the anterior segment of the eye, when ocular barriers are intact, remains a challenge. Although several methods and strategies have shown positive preclinical results, clinical translation remains limited by safety, tolerance, and acceptability, as well as by regulatory challenges. In pathologic conditions, when ocular barriers are disrupted, the local delivery of gene therapy products remains difficult due to enhanced inflammatory environment, poor predictability of pharmacokinetics, and increased systemic passage.

### 3.4. The Barriers in the Posterior Segment

#### 3.4.1. The Choroid

The choroid is formed by an organized vascular network including arteries, veins, and capillaries (choriocapillaries), as well as immune cells such as mast cells and microglial cells, melanocytes, fibroblasts, smooth muscle cells, and a dense network of sympathetic and parasympathetic fibers. The choroidal vessels are part of the external blood–retinal barrier, protecting the retina from substances circulating in choroidal vessels, whose blood flow is among the highest of any organ.

The vessels of the human choriocapillary are constituted by endothelial cells with tight junctions, but provided with large diaphragmatic fenestrations. Such fenestrations can also be found in the kidney, for example, but they are only diaphragmed during development and not in adults. Unlike the other fenestrated endothelia found in vascular beds, these diaphragms considerably limit the passage of molecules of high molecular weight (Einstein–Stokes radius > 3.2 nm). Proteins can be retained not only because of their size, but also by electrostatic repulsion due to the negative charge and high anionic density of the capillary endothelium surface [46]. The fenestrations form pores of 60 to 80 nm in diameter, but the cavities formed by the diaphragm reduce this size about 10-fold. The number of fenestrations and the opening of the diaphragms are under the regulation of VEGF, which ensures the production and function of the plasmalemma vesicle-associated protein (PLVAP), an essential constituent of fenestrations [47,48]. Other transport systems allow the controlled passage of proteins from the vascular compartment to the retinal pigment epithelium (RPE), particularly caveolae, which are 60–80 nm in diameter and are responsible for the passage of albumin and other macromolecules, vesiculo-vacuolar organelles (0.12–0.14 mm), and transendothelial channels, which are thought to be assemblies of caveolae. It has been shown that caveolae ensure, at least in part, the passage of albumin and all the molecules bound to it, from the capillary choroidal endothelium to the RPE.

#### 3.4.2. Bruch’s Membrane (BM)

BM is constituted by five layers. From the RPE to the choroid, the specified layers can be identified histologically as the basal lamina of the RPE, the inner collagen layer, the elastin layer, the outer collagen layer, and the basal lamina of the choriocapillaries.

BM behaves as a semipermeable, passive, acellular filter for the reciprocal exchange of biomolecules between the retina and the choroid. Diffusion through the BM varies according to its molecular composition, which is affected by numerous factors such as aging and location in the retina. The elastic layer has the largest pore size and the greatest water conductivity, while the inner collagen layer has the smallest pores and the lowest conductivity. The water permeability of the BM is impacted by age-related collagen crosslinking and by the accumulation of hydrophobic lipids (lipid wall) and membrane debris that increase with age [49].

Diffusion across the BM also relies on hydrostatic pressure on both sides of the BM, as well as the maintenance of gradients of specific biomolecules and inorganic ions. The structure and the molecular composition of the BM vary between the macula and the periphery. The permeability of the BM to proteins up to 150 kDa molecular weight has been demonstrated [50]. However, in humans, the permeability of the BM to plasma proteins decreases with age, with a remarkable linear association between the permeability to proteins of lower molecular weight and age. The basal hydraulic conductivity of BM declines exponentially with age with a half-life of 19 years. In addition, this reduction in permeability is significantly greater in the macula, reaching up to 90% for macromolecules of 20 kDa. In patients with AMD, the limitation of transport may be further increased [22,51]. The BM also limits the transport of lipophilic solutes, notably cationic solutes, compared to hydrophilic solutes. The reduction in transport across this physiological barrier is directly related to the binding of the solute to the tissue [52]. Thus, the penetration of high-molecular-weight molecules, such as anti-VEGF antibodies, may be reduced in the oldest subjects with AMD.

#### 3.4.3. The Retinal Pigment Epithelium (RPE)

The RPE is considered as the only component of the outer blood–retinal barrier because of its tight junctions (Figure 3). However, the RPE is one of the constituents of the outer barrier that includes the choriocapillary vessels, the BM, and the outer limiting membrane. The resistance of the tight junctions of adult human RPE is approximately 40 Ω·cm^2^, which makes RPE a low-resistance epithelium. The RPE does not undergo mitosis in a physiological situation (this point is still in debate), and a loss of about 2% of the RPE per decade of life is measured without a decrease in the foveolar region due to a probable reorganization and migration from the periphery to the center. The RPE plays a major role in controlling potassium concentrations in the subretinal space during phototransduction through the expression of apical pumps (Na^+^/K^+^-ATPase) and basolateral Na^+^/K^+^/2Cl^−^ co-transporter. It ensures the regulation of subretinal volume by actively removing water from the subretinal space at the rate of several µL/cm^2^/h, due to the presence of transmembrane proteins forming water channels (aquaporin 1) and the outgoing gradient of Cl^−^. This ionic movement is supplied by the Na^+^ gradient which is maintained by the Na^+^/K^+^-ATPase electrogenic pump. Chloride (Cl^−^) ions exit through the basolateral membrane via chloride conductance modulated by intracellular calcium concentration. The transport of bicarbonate (HCO_3_^−^) can also boost the elimination of liquids. The efflux of HCO_3_^−^ through the basolateral membrane is accompanied by fluid transport from the subretinal space to the choroid.

Proteins do not cross an intact RPE intercellularly [53], but albumin may enter the RPE through vesicular transports dependent on caveolin. Numerous studies have investigated the transepithelial penetration of proteins that pass through the sclera, such as pigment epithelium-derived factor (PEDF, 50 kDa), ovalbumin (45 kDa), or soluble vascular endothelial growth factor receptor I (VEGF) (VEGFR-I or sFlt-1, 110 kD), showing that the RPE constitutes a barrier to the penetration of high-molecular-weight proteins. Using fluorescein isothiocyanate (FITC) labeled 4–80 kDa dextrans, the entering and outgoing permeability of the RPE was shown to similarly and exponentially decrease with increasing molecular radius. In general, 50–75 kD proteins can enter in the choroid, which is not the case for the retina [7] (Figure 3A and Figure 4).

Previous work established the bidirectional permeability of eight low-molecular-weight drugs and bevacizumab (approximately 149 kDa) through isolated bovine RPE choroid. The permeability of small molecules is 10^−6^–10^−5^ cm/s. Most low-weight drugs pass from the apical to the basal side and vice versa. However, most hydrophilic molecules have a permeability 5– times lower compared to the most hydrophobic molecules. Ciprofloxacin and ketorolac show increased passage out to the basolateral surface (six and 14 times greater compared to when directed inward, respectively), which could indicate active transport. In contrast, only 3% of the intravitreal clearance of bevacizumab is explained by a transepithelial passage, confirming the weakness of protein transport through the RPE [54]. Under pathological conditions or during aging, the tight junctions of the RPE cells can be impaired, which considerably alters the circulation of proteins through the epithelial layer. In a prematurely senescent human RPE cell line, the expression of junction proteins was altered and the permeability of the RPE was increased [55]. Placental growth factor can disrupt the tight junctions of RPE cells through VEGFR1 [56]; in diabetes, the RPE junctions are also altered [57]. Therefore, under different pathological conditions, the focal disruption of the RPE can modify the passage of macromolecules through the barrier of this epithelium. The existence of specific properties or dynamics of tight junctions at the macular level has not been studied. The RPE also constitutes the central site of xenobiotic metabolism in the eye. Lysosomal enzymes in lysosomes and melanosomes can cause hydrolysis of various proteins and generate the formation of degradation products of lower molecular weight in an acidic environment. It is possible that this activity also decreases during aging.

Nevertheless, the suprachoroidal route has been shown to allow the delivery of viral vectors and of plasmid DNA to the RPE cells.

#### 3.4.4. The Retina

In the retina, the outer limiting membrane (OLM), plexiform layers, and the inner limiting membrane (ILM) are obstacles to the free diffusion of molecules (Figure 3 and Figure 4). According to their molecular weight, conformation, and charge, proteins could undergo limited lateral migration.

##### The Outer Limiting Membrane and the Plexiform Layers

In the vertebrate retina, the apical ends of Müller’s glial cells (RMGs) are attached to each other and to the inner segments of photoreceptors by tight heterotypic junctions, particularly between cones and RMGs, which constitute the OLM [58] (Figure 3B and Figure 4). The OLM forms a selective barrier to the diffusion of proteins >35 Å in diameter, which corresponds to the diameter of albumin. In retinal detachment models, the diffusion of molecules from the subretinal space to the vitreous is slowed beyond this size [53,59] with high molecular-weight proteins that remain under the retina for several days.

The plexiform layers also form barriers to the diffusion of protein and water across the retina. Using Ussing’s chamber, the permeability limit of the retina to high-molecular-weight compounds was measured around 80 kD in both humans, pigs, beef, and rabbits (Figure 4). The resistances were higher in the inner (IPL) and outer (OPL) plexiform layers in accordance with previous works in monkeys, showing that OPL constituted a barrier to the diffusion of peroxidase [60].

##### The Inner Limiting Membrane (ILM)

The ILM is constituted by the basal lamina of RMGs. In the retinas of monkeys and humans, the ILM consists of three distinct strata, the lamina rara interna, immediately adjacent to the terminal feet of RMGs, the lamina densa, and the lamina rara externa, which is contiguous with the vitreous cortex. The ILM morphology is topographically different in the primate retina; the lamina densa of the peripheral retina is thin, but it becomes increasingly thick and convoluted in the posterior retina, except along the fovea and where large vessels juxtapose with the inner surface of the retina. In both monkeys and humans, ILM thickens during aging.

Conflicting results have been published regarding the interfering role of ILM in protein diffusion following intravitreal injection (IVT). While studies have shown that proteins with a molecular weight as high as 150 kDa are crossing the ILM, others indicate that the ILM restrains the passage of substances larger than 4.5 nm in molecular radius, i.e., molecules larger than 40–70 kDa, depending on their shape and Einstein–Stokes radius (Figure 4).

Likewise, recombinant humanized monoclonal antibodies labeled with ^125^I (rhuMAb) against human epidermal growth factor receptor 2 (HER2) (rhuMAb HER2, 148 kDa) and Fab fragments of rhuMAb against VEGF (rhuMAb VEGF Fab, 48 kDa) were injected intravitreally into rhesus macaques. Whereas rhuMAb VEGF Fab was evenly distributed throughout all layers of the retina, rhuMAb HER2 accumulated at the ILM and did not reach the deeper layers of the retina regardless of the time of the exam [61]. These studies formed the basis for the development of specific anti-VEGF Fabs of a lower molecular weight. However, other reports have shown that bevacizumab (149 kDa; Avastin^®^, Genentech, South San Francisco, CA, USA), a complete rhuMab directed against human VEGF, could diffuse into all layers of the retina following intravitreal injection in different species, including monkey. In albino rabbits, 24 h after intravitreal injection of bevacizumab (2.5 mg in 0.1 mL) [62], bevacizumab was detected in all layers of the retina and in the subretinal space [63]. In monkeys, bevacizumab initially localized in the inner layers of the retina, before accumulating in the outer segments of the photoreceptors at 1 week and up to 14 days, being at least partly mediated by RMGs [64]. In the fovea, bevacizumab accumulates preferentially at 24 h after intravitreal injection [65], which suggests that the penetration is different in the macula compared to other areas of the retina. More recently, it was shown that CNTF (23 kDa) penetrates very weakly into the outermost retinal layers through the ILM [66]. The retinal penetration of adeno-associated virus (AAV) vectors was also studied after intravitreal injection. While it was possible to transfect retinal cells after intravitreal injection of rAAV2 in mice, this administration was ineffective in rats due to the presence of the ILM [67] lacking the adhesion receptor. In contrast, the digestion (even partial) of ILM by proteases increases the penetration and transfection of retinal cells after administration of certain types of AAV. AAV 5 for instance, which does not adhere to the intact ILM, can enter the retina if the ILM is lacking or damaged [68]. Transretinal migration of proteins may not only depend on weight and molecular structures, but also be regulated by active and specific transports from RMG cells. The existence of specific receptors and/or efflux proteins at the level of the feet of RMG cells remains to be demonstrated. Techniques are currently aimed at combining the intravitreal injection of viral vectors after removing the ILM by peeling [69] or with an injection under the ILM [70].

As the ILM exhibits specific structural features at the fovea level, studies concerning the penetration of different molecules assessed in rodents or lagomorphs cannot be extrapolated to the human eye.

#### 3.4.5. The Inner Blood–Retinal Barrier

The inner blood–retinal barrier (IWRB) consists of endothelial tight junctions, densely packed pericytes (one pericyte to one endothelial cell), astrocytes, and the terminal feet of RMG cells that surround the retinal capillaries (Figure 3D). All these structures are necessary for the formation and maintenance of the BHRI. Tight junctions, only present in the endothelia of retinal and brain vessels, are composed by a complex of belt-like tight junctions, forming a trans-endothelial resistance of approximately 1500–2000 Ω·cm^2^, which is comparable to the blood–brain barrier. These junctions contain predominantly claudins 5, 1, and 2, JAM-A, and ZO-1, 2, and 3 proteins. Endothelial cells are not windowed under physiological conditions, and capillaries are impermeable to proteins, which can nevertheless cross their membrane by transcytosis, especially during development. VEGF increases endothelial permeability, mainly by increasing transcytosis, but it can also induce junctional protein phosphorylation, which induces their nuclear translocation. Activation of a Rho-GTPase family kinase disorganizes the cytoskeleton, including the junctions, via a constriction mechanism of the actin network. Proinflammatory cytokines such as IL-1, IL-6, TNF-alpha, ICAM-1, chronic hyperglycemia, complement activation, and oxidative stress also weaken the inner BHR barrier contributing to the macular edema formation [71].

#### 3.4.6. The Vitreous

The vitreous is predominantly water (98%) containing a low concentration of soluble proteins, ions and low-molecular-weight solutes. The two main structural components are collagen (40–120 μg/mL) and hyaluronic acid (100–400 μg/mL). However, it also contains chondroitin sulfate and possibly heparan sulfate. The vitreous is a complex three-dimensional network that can particularly restrict the diffusion of large molecules. In vitro, the vitreous reduces the cellular uptake of FITC-dextrans from 44 kDa to 77 kDa by 65% to 100%. In vivo, the mobility of large molecules, especially positively charged, would theoretically be limited within the vitreous humor. However, convective diffusion resulting from eye movements accounts for about 30% of total intravitreal drug transport in humans. This effect may be even greater for higher-molecular-weight compounds, which diffuse more slowly [72]. The water flow from the anterior vitreous to the posterior vitreous must also be considered. Using a computer model developed to describe the three-dimensional convective–diffusive transport of a drug released from a controlled-release intravitreal source, it has been shown that the ratio of the drug amount reaching the retina to the amount of drug removed by the aqueous humor is 2.4 for small molecules but 13 for large molecules [73]. Thus, a large proportion of macromolecules are eliminated by the anterior route.

During aging and under pathological conditions, as well as after repeated IVTs, the vitreous structure can be disorganized and significantly alter the diffusion of injected compounds from the injection site to the retina. Considering that macromolecules, depending on their charges, may not diffuse freely into the vitreous, the site of the needle during injection may influence the bioavailability of the injected active protein targeting retinal cells.

### 3.5. Macula Specificities

The inner limiting membrane thickness increases from the periphery, where it measures about 400 nm, to the macula where it reaches 2 to 4 µm. However, at the fovea, the inner limiting membrane thickness is greatly reduced around 40 to 100 nm. In front of the macula, the vitreous forms a space called the “pre-macular bursa”. This can be described as a muffle, with the part of the thumb located at the beginning of Cloquet’s canal on the optic nerve, the part of the finger on the macula, and the wrist and arm making their way toward the front of the eye, ending behind the lens. Thus, there would be a true “passage” possible from the front of the eye to the macular region through this space.

Functional macular specialization is associated with anatomical specificities: high density of cones, central avascular zone, and centrifugal displacement of cone and Müller cell axons that reach several hundred micrometers following a “Z”-shaped trajectory. Along this trajectory, photoreceptor axons and Müller cells are associated with junctional proteins such as ZO-1 or claudin 5, preventing protein accumulation in this specific region [74,75]. In addition, AQP4 is localized in the perivascular tips of astrocytes, as well as around Müller cells in the macula, and it provides low-resistance pathways for fluid movement between the paravascular spaces and the interstitium. This creates a “lymphatic-like system” for drainage of interstitial solutes and proteins [71].

These observations might suggest a permanent flow of dissolved substances created by AQP4 channels. This would form a hydraulic pathway along the RMG cells in the foveal fossa, through Henle’s fibers, to the optic nerve head. At the optic nerve head, previous studies have identified a defect in the blood–retinal barrier integrity, allowing the drainage of retinal protein [71]. These drainage pathways may have important pharmacokinetic implications under normal or pathological conditions.

### 3.6. Impact of Ocular Barriers on Nucleic Acid and Gene Delivery to Tissues and Cells of the Posterior Segment of the Eye

The systemic route is not suitable for the administration of gene therapy products for the treatment of posterior segment diseases of the eye, unless manipulation is applied to increase the blood barrier permeability. Modulation of the blood–brain and retinal barriers using RNAi-based methods for suppression of claudin-5 was used to transiently increase their permeability, but this was limited to molecules smaller than 1 kD [76]. The intravitreal route, on the other hand, has clearly emerged as a preferred route because the ocular barrier contains the injected products in situ. However, once in the vitreous cavity, naked nucleic acids and plasmids, viral vectors, and nonviral polyhedral and lipid vectors do not simply reach the targeted retinal cells. Physicochemical interactions with vitreous molecules, internal and external limiting barriers, the cell membrane-rich structure of the retinal layers, and their organization limit simple diffusion to photoreceptors and pigment epithelium cells. Thus, once injected into the vitreous, we and others showed more than 15 years ago that naked oligonucleotides readily penetrate the inner ganglion and nuclear layers of the retina, but poorly penetrate the photoreceptors and even more poorly penetrate the RPE cells [77,78]. Transpalpebral or transscleral iontophoresis was shown to enhance the photoreceptor delivery of oligonucleotides injected into the vitreous through modulation of the transcellular pathway in retinal glial Müller cells. This method was used for gene correction in photoreceptors for the treatment of retinitis pigmentosa [77,79], but with a low correction rate. Yet, several RNA-based therapies, administered intravitreally, are currently under development to treat inherited retinal degeneration, using siRNA, shRNA for autosomal dominant diseases, or antisense oligonucleotides which can restore splicing defects [80]. Using a high dose of RNA molecules, sufficient drug levels might be reached in the photoreceptors. Other attempts to deliver nucleic acids or plasmids encapsulated in particulate systems, after intravitreous injection, have been shown to reach the RPE cells through the transcellular pathway of Müller cells [37,81,82,83]. Inducible systems can also be used, such as light-sensitive peptidic nanoparticles formed by VP-22 cationic peptide and oligonucleotides, which, after intravitreous injection penetrated in retinal cells and RPE cells, stayed stable for several weeks, and released the active oligonucleotide upon laser or transccleral illumination [83]. However, to date, manufacturing reproducibility, sterilization processes, and the proinflammatory potential of polymers have limited the clinical translation of these strategies. Various serotypes of viral vector such as AAV or lentiviruses are being tested to transduce different types of retinal cells located either in the superficial or in the deeper retinal layers, in order to avoid the subretinal administration, which allows the proper diffusion of viral vectors and the efficient transfection of photoreceptor cells and RPE cells within, or at the proximity of the injection site, where the bubble of subretinal detachment has formed [44,84]. Excellent and extensive reviews on different viral vectors and their clinical use have recently been published [85,86,87,88]. The regionalization of the viral transfection efficacy, restricted to the area of retinal detachment where the virus has been administered, demonstrates that simple diffusion within retinal layers and in the subretinal space is hampered by dynamic barriers.

To target the RPE cells, but more specifically the choroidal endothelial cells, that do proliferate in case of choroidal neovascularization, the injection of gene therapy products into the supra choroidal space is more adequate, allowing to reach the target cells without the need for transretinal or subretinal passage. The first attempt was made by suprachoroidal injection of plasmid DNA followed by electrotransfection. Efficient transfection of RPE and choroidal cells, as well as sustained production of sFlt-1, was achieved for several months, preventing the development of laser-induced choroidal neovascularization [89,90,91]. Years later, suprachoroidal administration of viral vector was proposed to deliver antiangiogenic proteins to treat choroidal neovascularization [92], but with the risk of systemic diffusion.

In summary, there is no doubt that a major advantage of the ocular barriers is that they prevent the uncontrolled systemic dispersion of gene therapy products administered into the eye. Yet, diffusion through the optic nerves to the other eye and to the brain cannot be excluded and has been advocated as a potential explanation for the bilateral effects of lenadogene nolparvovec, an AAV-2 serotype 2, containing a codon-optimized complementary DNA encoding the human wildtype MT-ND4 subunit protein, injected into the vitreous of one eye of patients with Leber hereditary optic neuropathy, resulting in a clinical effect in the contralateral, noninjected eye [93]. It is not fully understood how, but clinical studies have shown that intravitreous injection of viral vectors can elicit an inflammatory and immune response with potential deleterious consequences, which has reduced the enthusiasm for intravitreous viral vector delivery [94,95]. Gene delivery and oligonucleotide delivery to retinal cells remains a challenge. A better understanding of the barriers that limit the penetration of the different vectors into the retina and into the retinal cells, as well as of the fate of the different vectors in the short and long term, is still needed for progress to clinical use.

## 4. Concluding Remarks and Future Directions

Ocular barriers are essential to maintain ocular environments and intraocular microenvironments. Despite numerous studies allowing a better understanding of the functioning, the structure, and the regulation of barriers, much remains to be discovered. The barriers and their properties are modified during development, aging, and various pathological processes, even if they are not clinically detectable using the usual methods of exploration. These modifications have definite, sometimes ignored, often underestimated consequences for ocular pharmacokinetics and drug bioavailability. This should encourage kinetic studies in pathological models and in humans.

The eye offers the major advantage of being directly accessible and directly observable with improved and optimized visualization methods. Technological and surgical developments have prompted a number of clinical studies testing gene therapy products using viral vectors, as well as nonviral methods and even naked nucleic acids. Unfortunately, the rate of success remains very low, and failure to reach a clinical endpoint raises the crucial question of whether, in humans, the gene therapy product can reach its target at the efficient dose. This question cannot be answered with certitude taking into account that ocular pharmacokinetic studies cannot be performed easily in patients and that gene therapy products do not follow classical pharmacokinetic models. Most preclinical studies are conducted in rodent models of ocular diseases and distribution studies conducted in larger eye animal models such as rabbits or nonhuman primates. However, translation from efficacy to distribution and from animal models to human diseased eyes remains uncertain and speculative. There is a crucial need to continue and study how endogenous molecules, including antigens, are transported across the ocular tissues and the ocular barriers, to better anticipate the behavior of gene therapy products in normal and pathological conditions.

Targeted delivery of gene therapy products at therapeutic levels, without overdosing, at the site of pathological process, in the appropriate cell, is the key for success. This will be possible through a combination of technologies, including the vector itself, the local administration technique, visualization and guided procedures, and the control of the administered dose. However, to be adopted by practicians and regulatory authorities, the combination therapy product must still remain simple to use and manufacturable for clinical use.

## Figures and Tables

**Figure 1 pharmaceutics-14-00998-f001:**
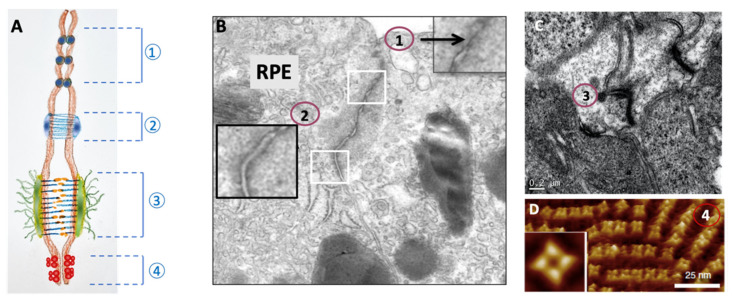
(**A**) Schematic representation of the different types of junctions allowing contact between cells (zonula occludens or tight junction (1), zonula adeherens (2), macula adherens, or desmosomes (3) and gap junctions or connexon (4)). (**B**) Microphotographs of retinal pigment epithelium (RPE) using transmission electron microscopy images of tight junction (1) and zonula adherens (2). (**C**) Microphotograph of glial Müller cells at the outer limiting membrane showing desmosome (3). (**D**) Atomic force micrograph of the connexons formed by AQP0 in the lens (4).

**Figure 2 pharmaceutics-14-00998-f002:**
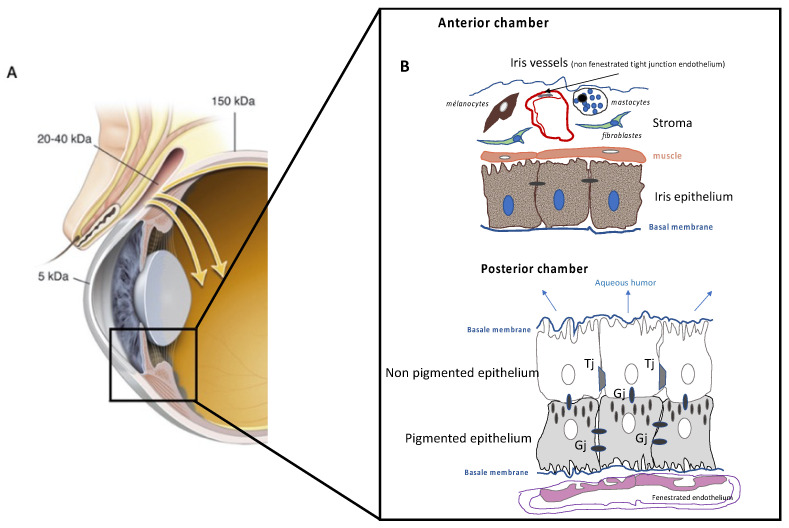
Hemato-aqueous barrier. (**A**) Macroscopic representation of the anterior part of the ocular globe. The molecular weights indicate the size limit for molecular passage through the outermost tissues (conjunctiva, cornea, anterior sclera). Inset: location of the two major components of the blood–aqueous barriers are identified as iris vessels (1) and nonpigmented ciliary body epithelia (2). (**B**) Schematic representation of the components of the blood–aqueous barrier in the iris where all cell components are represented with the iris vessels and its tight-junction endothelial cells (1) (upper image), as well as the ciliary body, with all its cellular components and its tight-junction nonpigmented epithelium (2) (lower image). TJ: tight junctions, GJ: gap junctions.

**Figure 3 pharmaceutics-14-00998-f003:**
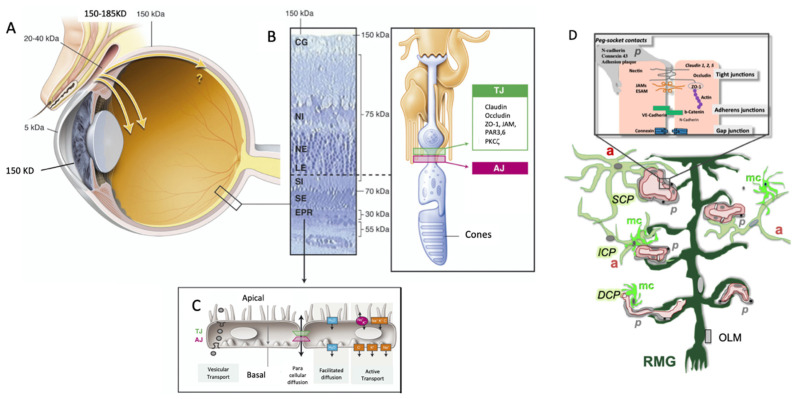
(**A**) Schematic representation of the ocular globe. (**B**) Retinal pigment epithelium and representation of the different transport mechanisms. TJ: tight junction, AJ: adherent junction. (**C**) Histologic section of the peripheral retina with the different layers (GC: ganglion cell layer, INL: inner nuclear layer, ONL: outer nuclear layer, OLM: outer limiting membrane, IS: inner segment of photoreceptors, OS: outer segment of photoreceptors, RPE: retinal pigment epithelium) and molecular weight limits for molecule passage. Inset: constituents of the OLM, formed by junctions between photoreceptors and retinal glial Müller cells (RMGs). TJ: tight junction, AJ: adherent junction. (**D**) Components of the inner hemato-retinal barrier, with the tight junction endothelial cells, peg–socket contact with pericytes (p), and links with astrocytes (a) in the superficial capillary plexus (SCP) and the intermediate capillary plexus (ICP), with RMG cells all long the retinal layers, and with microglial cells (mc).

**Figure 4 pharmaceutics-14-00998-f004:**
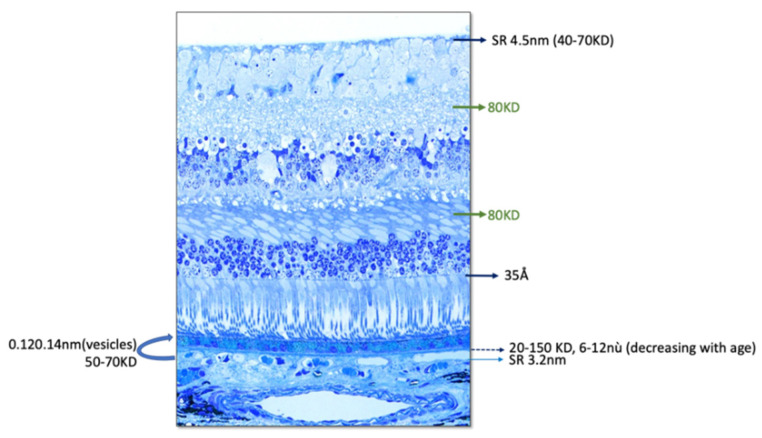
Semi-thin section of human retina with indication of known size and molecular weight limits of the different layers and tissue structures (SR = Stokes radius).

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
