# Peer review of "Ocular Barriers and Their Influence on Gene Therapy Products Delivery"

_pharmaceutics, 2022, doi:10.3390/pharmaceutics14050998_

Round 1
Reviewer 1 Report
The manuscript discusses various physiological and anatomical ocular barriers and their influence on drug delivery. It is well written overall. However, the authors are requested to review the comments below and address them appropriately:
There are several grammatical and syntax errors in the manuscript that authors need to rectify.
The layout and the depiction of barriers in Figure 1 on page 2 appears unclear. Not sure why the junction proteins and the transporters efflux proteins are included within a circle and the other components around the circle. The figure needs to be structured appropriately.
It is better to avoid ellipsis in your sentences, such as, in line 5 of section 2.
Page 3 – last line: You refer to Figure 1a and b, however, should this read as Figure 2a and b instead? Likewise, in the last line of section 2.5, Figure 1b,4 should read as Figure 2b,4?
Figure 2 – need to clearly identify on the figure which parts are ‘A’ and ‘B’.
Figure 2 legend on page 4: ‘tight’ in fourth line should read as ‘tight junction’. Fifth line states …desmosomes (b) and ….in the lens (c). However, should these read as ….desmosomes (3) and ….in the lens (4)?
Page 5 and lines 7 and 10: ‘Figure 2’ and ‘Figure 2b’ cited appear to be missing.
Page 5 bottom two lines – Please remove epllipsis.
Section 3.1.2 to section 3.2 – There is an extensive narrative on the anatomy of each barrier. However, including examples of drugs that are affected by these barriers and transport mechanisms would be useful and interesting, in addition to adding proper context to the discussion.
Page 6 – second line from the bottom: You state “see chapter on pharmacokinetics”. However, this is missing in the manuscript. Please check.
Page 7 – second line: you state “particulate formulations” for transcorneal absorption but there is no discussion on how these systems help. This needs further discussion.
Page 7 – fourth line from the bottom: “Figure 4 and 5” stated there appear to be missing.
Page 8 – Figure 3B shown in the box is an enlarged image of ‘Figure 3A. However, the image does not clearly show the individual locations of hemato ocular barrier in the iris and the ciliary body.
Page 9 – Section 3.3.2. heading: “The Bruch’s membrane (MB)”, the abbreviation in the bracket should read as ‘(BM)’.
Page 9 – 5th line from the bottom: “The RPE does not divide in a physiological situation”. This statement is unclear and needs further explanation.
Page 13 – Figure 4D has labels that are too small to read.
Alongside conclusions, it would be useful to include a paragraph on the ‘future perspectives’ on emerging science and technology in overcoming these barriers in drug delivery.
Reviewer 2 Report
The review manuscript written very well. The manuscript written with of proper flow and the manuscript is discussing everything in detail.
- Write the role of efflux transporters in the treatment of ocular tumor treatment.
- Effect of barriers on tear kinetics from novel delivery systems, and their influence further on the back of the the eye delivery.
- Write the latest delivery approaches for the improved ocular delivery with involvement of ocular barriers.
- Write the effect significance of barriers for the delivery through various routes of administration.
Reviewer 3 Report
In this manuscript, the authors reviewed the Ocular barriers and their influence on drug delivery. In my opinion, some issues should be further addressed and I hope the following comments could be helpful for improving their paper.
- In the introduction, the background about drug delivery is little, the authors should enrich this part and emphasize the necessity of "Ocular barriers and their influence "
- Authors focused on Ocular barriers, but what are the distinguished properties and specific problems of drug delivery? The authors never discussed it.
- Good quality figures are very important for a good review paper, Try to add at least 4-5 figures more in this manuscript from recent literature and also improve the quality of figures.
- The authors should summarize the current approaches to fabricating "ocular drug delivery" and compare their advantages and disadvantages in order to draw the reader's attention.
- This manuscript is well organized but lacks specific comparative analysis. What are the advantages of "ocular drug delivery" compared with traditional technology?
- Please revisit the entire manuscript for minor grammar issues.
- In conclusions and perspectives, the author should consider giving some methodological design about how to improve the performance of ocular drug delivery in future.
- Kindly summarize overall section 3. Description of ocular barriers in one table.
- Future perspective is very important for review paper, kindly add this section.
Round 2
Reviewer 2 Report
Manuscript modified as per the suggestions.
This manuscript is a resubmission of an earlier submission. The following is a list of the peer review reports and author responses from that submission.